# Non-Coding RNAs in Retinoic Acid as Differentiation and Disease Drivers

**DOI:** 10.3390/ncrna7010013

**Published:** 2021-02-17

**Authors:** Carlos García-Padilla, Estefanía Lozano-Velasco, Carmen López-Sánchez, Virginio Garcia-Martínez, Amelia Aranega, Diego Franco

**Affiliations:** 1Department of Experimental Biology, University of Jaen, 23071 Jaen, Spain; carlosgp@unex.es (C.G.-P.); evelasco@ujaen.es (E.L.-V.); aaranega@ujaen.es (A.A.); 2Department of Human Anatomy and Embryology, University of Extremadura, 06006 Badajoz, Spain; clopez@unex.es (C.L.-S.); virginio@unex.es (V.G.-M.); 3Institute of Molecular Pathology Biomarkers, University of Extremadura, 06006 Badajoz, Spain; 4Fundación Medina, 18016 Granada, Spain

**Keywords:** retinoic acid, microRNAs, lncRNAs, development, disease

## Abstract

All-trans retinoic acid (RA) is the most active metabolite of vitamin A. Several studies have described a pivotal role for RA signalling in different biological processes such as cell growth and differentiation, embryonic development and organogenesis. Since RA signalling is highly dose-dependent, a fine-tuning regulatory mechanism is required. Thus, RA signalling deregulation has a major impact, both in development and disease, related in many cases to oncogenic processes. In this review, we focus on the impact of ncRNA post-transcriptional regulatory mechanisms, especially those of microRNAs and lncRNAs, in RA signalling pathways during differentiation and disease.

## 1. Retinoic Acid Biogenesis

All-trans retinoic acid (RA) is a major active metabolite of vitamin A [1]. It has long been recognised that this fat-soluble vitamin plays essential roles throughout all the stages of development [2]. Many reports have described a pivotal modulatory function for RA signalling, in a dose-dependent manner, in different biological processes, including cell growth and differentiation, early embryonic development, organogenesis and cell homeostasis in diverse tissues and systems [3,4,5]. Additionally, 9-cis retinoic acid (9C-RAs), another metabolite derived from vitamin A, has been reported [6,7], although its role appears to be restricted to adult pancreas, requiring further investigation in other biological contexts.

In most species, vitamin A is obtained from diet. Digestion and absorption of this vitamin require successive enzymatic reactions carried out by pancreatic enzymes that catalyse its conversion into retinol, following which it circulates in the blood bound to a specific transporter protein called RBP4 [8] (Figure 1). The interaction between RBP4 and its transmembrane receptor, STRAT6, allows retinol to enter the intracellular space [9]. In the cytoplasm, retinol is subjected to two consecutive oxidative reactions, RA being their final product. In the first oxidative step, retinol is processed to retinaldehyde by the bivalent RDH10/DSH3 enzyme complex [10,11,12] and, in the second phase, RALDH dehydrogenases, particularly RALDH2, catalyse oxidation of retinaldehyde into RA [13,14] (Figure 1).

RA enters the nucleus in association with an intracytoplasmic transporter, CRAPBII, which releases it directly to specific nuclear receptors, RARs [15,16,17] (Figure 1). Another isoform of this enzyme, CRAPBI, can also modulate RA input into the nucleus, although its response efficacy is lower. Furthermore, RA transported by CRAPBI is released in the nuclear periphery and not into the RARs’ proximity [18,19,20]. These RAR nuclear receptors form transcriptional heterodimers with another class of nuclear receptors, RXRs (Figure 1). The role of both receptors has been extensively studied using genetically engineered deficient animal models, which have highlighted a pleiotropic and redundant function for the different receptors that is in fact, in most cases, evolutionarily conserved [20,21,22,23,24]. The heterodimers of RARs/RXRs recognise RARE motifs, characterised by the 5′-PuG (G/T) TCA-3′ repetitive DNA motif, promoting transcription of multiple target genes involved in cell growth and differentiation, development and/or metabolism [25,26] (Figure 1). However, the RA signalling pathway does not only act through transcriptional activation. Notably, non-transcriptional effects of RA and RARs have recently been described, especially on the activation of MAPK kinases [27,28,29]. In this line of research, novel evidence has also demonstrated that, in elevated RA cytoplasmic levels, CRAPBI constitutes a protein complex with RAF/MEK, modulating Erk 1/2 phosphorylation, which is then capable of promoting nuclear entrance and thus the arrest of cell growth by p27 phosphorylation [30].

Since RA signalling is highly dose-dependent, a fine-tuning regulatory mechanism capable of maintaining proper RA intracytoplasmic levels is needed [31]. RA levels are generally regulated by means of both degradation and synthesis. On the one hand, CYP26 enzyme family members, and particularly CYP26A1, degrade RAs into different metabolites, such as 4-OH-at-RA or 18-OH-at-RA [32,33,34,35]. These metabolites have the potential capacity to bind to RARs and may thus represent potent teratogens [36,37]. Detraction of RA by CYP26 enzymes results in ineffective activation of RARs/RXRs and thus downregulation of RA signalling. On the other hand, RA synthesis is modulated by RALDH family enzymes, mainly RALDH2, as described in greater detail above. A proper balance between CYP26A1 and RALDH2 expression is necessary to establish the adequate RA dose [38]. In fact, many studies have reported that altered RA gradients during embryogenesis are translated into congenital defects and foetal malformations. Furthermore, gain and loss function assays have showed that RA signalling requires particular cellular RA levels to properly modulate RA-derived intracellular pathways [39,40,41,42,43,44].

Thus, RA signalling can be considered as a tight dose-dependent intracellular complex pathway regulated at different levels. In this review, we focus on the impact of the post-transcriptional regulatory mechanisms of ncRNAs, especially microRNAs and lncRNAs, in the RA signalling pathway during cell differentiation and disease.

## 2. Classification and Function of ncRNAs

For decades, scientists have considered non-coding RNAs (ncRNAs) as a non-functional part of the genome, focusing their attention on coding RNA biology. The sequencing of the human genome, and later the ENCODE project, have shown that more than 80% of the genome is transcribed in some type of RNA. Interestingly, only 3% of this transcribed genome corresponds to coding RNAs, suggesting that ncRNAs are as or more important than coding RNAs [45,46]. Currently, it has been demonstrated that non-coding RNAs are essential for the regulation of cellular pathways and biological processes such as cell development, differentiation, growth, homeostasis and disease (see, for recent reviews, [47,48,49,50]).

NcRNAs can be classified as two different classes according to their length. Small non-coding RNAs, represented by RNA molecules smaller than 200 nucleotides, include microRNAs, snoRNAs, piRNAs and tRNAs. Long non-coding RNAs [51], i.e., RNA molecules with sequences longer than 200 nucleotides, include several subtypes such as intronic lncRNAs, enhancer lncRNAs, circular lncRNAs and intergenic lncRNAs [52]. 

MicroRNAs are on average 20–22 ribonucleotides in length and display the capacity to bind to the 3’ untranslated region of coding RNAs by complementary base pairing and thus promote their degradation and/or translational blockage. The role of microRNAs as post-transcriptional modulators has been widely described in multiple biological and cellular processes, including cell development, differentiation, growth and homeostasis [53,54]. 

On the other hand, lncRNAs are structurally similar to mRNAs since they are transcribed by RNA polymerase II and have the same typical post-transcriptional modifications, i.e., 5′ terminal cap and 3′ terminal poly (A); but, importantly, they lack the capacity to code proteins. Mechanistically, they can act as both transcriptional regulators, modulating nuclear gene expression in different ways—such as epigenetic landscape control, transcriptional complex scaffolding or as decoy molecules—or as post-transcriptional regulators modulating microRNA degradation, mRNA stability and/or protein translation. Unlike microRNAs, our understanding of the regulatory roles of lncRNAs is just starting to be addressed. It is worth mentioning that the wide variety of functions that lncRNAs can perform might thus be a reflection of the importance of this class of RNA in the regulation of multiple biological processes (see reviews in [55,56,57,58]).

In this review, we focus on the role of microRNAs and lncRNAs in RA signalling during cell differentiation and disease, with particular emphasis on their role in cancer.

## 3. Role of MicroRNAs as Differentiation Drivers Mediated by RA

The balance between cell proliferation and differentiation is vital for the correct development of tissues in the embryo. In this context, RA signalling pathway modulates differentiation of several cell lineages by regulating a multitude of modulators. Activation or inhibition of these modulators leads to transcription of tissue-specific genes necessary for cell differentiation and repression of proliferation-enhancing genes in most cases. The functions of the different protein coding modulators have been widely described (see reviews in [59,60,61,62,63]) but not the role of non-coding RNAs. In recent years, microRNAs have been identified as important modulators of RA signalling pathways—functioning in most cases as downstream modulators—in different tissues, such as stem cells; neuronal, haematopoietic and skeletal myoblast; and germline differentiation, as detailed below (Figure 2).

### 3.1. MicroRNAs Involved in RA Signalling That Mediate Cell Differentiation of Pluripotent Stem Cells

The differentiation capacity of pluripotent stem cells has been widely described. These can give rise to a multitude of cells depending on the signals or the biological environment in which they develop. Interestingly, the RA signalling pathway has been described as a pro-differentiating signal that drives these stem cells towards different cell progenitors, such as skeletal myoblasts or neuronal progenitors. In this context, several studies have shown that RA-mediated stem cell differentiation leads to differential regulation of several microRNAs, which in turn regulate the translation of genes involved in this process, pointing to microRNAs as pivotal modulators of embryonic stem cell (ESC) differentiation for both human and murine ESCs.

To date, only two microRNAs, namely miR-34 and miR-145, have been described in human stem cell differentiation. Both microRNAs are upregulated by RA signals through p53 acetylation by the CBP/p300 complex. RA-induced acetylated p53 promotes expression of miR-34a and miR-145, which, in turn, inhibit the translation of pluripotency markers, such as LIN28, OCT4, KlF4, and SOX2. Loss-of-function assays have showed that miR-34 is dispensable for hESC differentiation but not miR-145. However, depletion of both microRNAs, and not only miR-145, leads to delayed hESC differentiation, resulting in most hESCs being OCT4 positive and suggesting that both microRNAs co-modulate this process [64]. 

Unlike for hESCs, microarrays have been carried out to explore deregulated microRNAs in murine ESCs treated with RA. A subset of these RA-regulated microRNAs has been described in differentiation towards skeletal (miR-10 and miR-214) and neuronal progenitors (miR-219 and miR-125b). While microRNAs involved in skeletal muscle differentiation play a pivotal role in epigenomic regulation, neuronal differentiation-associated microRNAs have been linked to correct expression of neuronal marker genes. 

Epigenomic regulation plays a vital role in the proper timeline of differentiation by modulating the appropriate gene expression pattern. Defects in this regulation lead to incorrect gene pathway activation and therefore to inaccurate cellular differentiation. For example, miR-10a and miR-214 have been described as important effectors in the differentiation of skeletal myoblasts through the modulation of epigenomic context by RA. Huang et al. [65] pointed to miR-10a as the critical mediator of ESC differentiation into smooth muscle cells (SMCs) by RA administration in its repression of HDCA4 protein levels, since HDAC4 is a negative chromatin complex regulator of SMC differentiation. RA treatment of ESCs upregulates miR-10a expression through NF-Kappa β nuclear translocation. In the nucleus, NF-Kappa β positively modulates miR-10a expression, which in turn binds to HDCA4 3′UTR, repressing its mRNA levels [65].

Similarly to HDCA4, polycomb repressor complex 2 (PRC2) plays a vital role in RA-mediated ESC differentiation [66]. In the absence of RA, PRC2 binds to several RAR/RXR-dependent gene promoters, preventing their transcription. In the presence of RA, the Mediator complex phosphorylates the 482 tyrosine residue of the EZH2 subunit of the PRC2 complex, promoting its dissociation from occupied promoters and thus leading to cell differentiation. In this context, miR-214, induced by RA signalling, reduces EZH2 protein levels and thus promotes PRC2-repressed genes expression and cell differentiation. Also, undifferentiated C2C12 assays have shown that upregulation of miR-214 accelerates myoblast differentiation by repressing EZH2 [67].

Like skeletal myoblasts, neuronal progenitors can be differentiated from ESCs by RA administration. Array-based miRNA profiling on mouse ESCs after 48 h of RA treatment identified 324 deregulated miRNAs; 43 miRNAs were upregulated and 281 miRNAs were downregulated, supporting a relevant role of this class of RNAs in cellular differentiation. Delving into the specific roles of this subset of microRNAs, Wu et al. (2017) showed that RA-mediated upregulation of miR-219 is essential for the repression of two transcription factors, *Foxj3* and *Zbtb18*, which are indeed important repressors of neuronal differentiation. Furthermore, downregulation of *Foxj3* and *Zbtb18* by miR-219 promotes the activation of *Olig1, Zic5, Erbb2* and *Olig2*, which are necessary factors for neural differentiation, thus enhancing this process [68]. Similarly, treatment of the P19 embryonic pluripotent carcinoma cell line with RA induces the expression of 19 microRNAs: let7a/b, miR-9, miR-30, miR-98, miR-100, miR-103, miR-124, miR-125, miR-128, miR-135, miR-156 and miR-218 [69]. Loss-of-function assays of those microRNAs have highlighted the role of miR-125b in repression of *Smad2, Smad4* and *Stat3* transcription factors, which in turn avoid differentiation. Furthermore, miR-125b overexpression induces neuronal differentiation by repressing expression of those transcription factors, leading to upregulation of neuronal markers such as *Scnba, Ephb2, Kcnq2, Flna, Syn2* and *Nefm* [69].

In summary, a large body of evidence demonstrates that the key functional role of RA-regulated microRNAs in stem cell differentiation is the silencing of several inhibitory genes and thus the promotion of stem cell differentiation. 

### 3.2. MiRNAs in RA Signalling-Mediated Neuronal Cell Differentiation

NT2 cell cultures exposed to different concentrations of RA have demonstrated its importance as a critical mediator balancing proliferation and differentiation in neuronal progenitor cells. RA modulates the expression of different transcription factors that fundamentally mediate cell cycle blockage and of neuronal markers that promote neuronal differentiation [70]. Several studies have pointed out the importance of microRNAs in RA-mediated neuronal differentiation acting upstream (miR-124) or downstream (miR-9 and miR-103) of this signalling pathway.

Specifically, the role of miR-124 has been widely described in RA-mediated neuronal differentiation. MiR-124 overexpression is capable of inducing neuronal differentiation by negatively regulating RARG expression, since RARG is expressed in the undifferentiated cells and maintains their pluripotential state [71]. Importantly, RARG overexpression leads to differentiation of mesenchymal but not neuronal cells. miR-124 has also been described as a repressor of the *Hes1* transcription factor, which is necessary for keeping progenitor cells in an undifferentiated state [72]. Loss of functions of miR-124 in P19 cells leads to upregulation of the Hes1 protein but not mRNA levels, suggesting that miR-124 indirectly modulates *Hes1* expression. Curiously, miR-124 expression is not sufficient to induce neuronal differentiation in P19 cells [73]. 

Importantly, *Hes1* is upregulated by *Id2*, another repressor of neuronal cell fate [74,75]. *Id2* is recognised by two RA-upregulated microRNAs, miR-9 and miR-103, which negatively modulate its expression. Unlike most microRNAs, miR-9-driven Id2 downregulation depends on binding to the coding region sequence located on the first exon of *Id2* mRNA, while miR-103 action is mediated by binding to the *Id2* 3’UTR region. Repression of *Id2* by both microRNAs leads to cell proliferation blockage and to *Vfg* upregulation, a transcription factor that is negatively modulated by *Id2* and is required for neuronal progenitor cell differentiation [76]. Overall, these data unravelled the emerging functional role of RA-regulated microRNAs in neuronal differentiation.

### 3.3. MiRNAs Involved in RA Signalling-Mediated Haematopoietic Differentiation

The significance of RA signalling for the appropriate development of the immune system, particularly during myelocytic and granulocyte differentiation, has been described in the literature for some time [77]. Altered RA levels or dysfunction of RARs/RXRs receptors lead to uncontrolled proliferation of hematopoietic progenitors and blockage of cell differentiation. Granulocyte differentiation mediated by RA upregulates the expression of several miRNAs, such as miR-29a, miR-142 and miR-223. Those miRNAs act as positive effectors downstream of RA signalling by modulating differentiation repressive factors. While miR-29a reduces the *Cdk6* mRNA levels, miR-142 represses *Tab2* translation, promoting granulocyte differentiation and blocking proliferation of progenitors. Also, *Cctn2*, a cell cycle factor that is necessary for proliferation of myelocytic progenitor cells, is a common target of both miR-29a and miR-142, suggesting a synergistic effect [78]. On the other hand, miR-223 represses NFIA expression, which is responsible for controlling genes involved in cell growth control [79]. Thus, these studies highlight the functional role of RA-modulating microRNA expression, including important downstream effects in immune system development.

### 3.4. MiRNAs Involved in RA Signalling-Mediated Skeletal Myoblast Differentiation

Unlike the majority of cell lines described before, RA administration maintains myoblast cells in an undifferentiated state. In fact, pharmacologic or genetic inactivation of endogenous RARs/RXRs promote myoblast differentiation. RA signalling represses the expression of early and late muscle differentiation markers and promotes expression of myogenic specification genes [80,81]. However, continued upregulation of RA signalling leads to myoblast differentiation. Importantly, RA excess negatively modulates α-dystrobrevin (DTNA) expression via miR-27b upregulation. Downregulation of α-dystrobrevin is responsible for MyoG depletion, an essential late myogenic differentiation transcription factor. Thus, RA-induced miR-27b upregulation promotes myoblast differentiation [82].

### 3.5. MiRNAs Involved in RA Signalling-Mediated Spermatogonia Differentiation

Spermatogenesis is subject to tight gene control orchestrated by RA, which plays a key role in differentiating from A_Ia_ spermatogonia, or undifferentiated spermatogonia, to A_I_ spermatogonia, or differentiated spermatogonia [83]. In dietary vitamin-A-deficient animal models (VADs), the spermatogenic differentiation is blocked, supporting the suggestion that RA signalling is essential in this process [84]. RA-mediated induction promotes the expression of specific markers that are essential for differentiation from A_Ia_ to A_I_, such as *Stra8* or *Kit*, among others, as well as distinct microRNAs. To date, several microRNAs have been reported as effectors of spermatogonia differentiation mediated by RA, namely miR-7b, miR-146 and microRNA clusters miR-17-92 and miR-106b-25 [85,86,87]. For example, miR-7b promotes spermatogonia differentiation by repressing *Mycn*, *Ccnd1*, *Colla2* and *Lin28* expression. In fact, Lin28 overexpression is capable of blocking RA-mediated differentiation itself. Therefore miR-7b acts as a vital mediator of spermatogenesis, repressing expression of spermatogonia differentiation factors [85]. 

In contrast to miR-7b, functional studies on the roles of miR-17-92, miR-106b-25 and miR-146, respectively, have demonstrated their repressive roles in sperm differentiation. miR-17-92 and miR-106b-25 downregulate the expression of RA-induced genes necessary for differentiation, such as *Bim*, *Kit*, *Socs3* and *Stat3* [86]. Similarly, miR-146 represses the expression of the Med1 cofactor, necessary for the transcriptional function of retinoic acid receptors (RARs/RXRs) and the expression of *Stra8* and *Schlh2*, which positively modulate *Kit* expression [87]. These data therefore support the hypothesis concerning the key role of RA-regulated microRNAs in germline differentiation. 

## 4. The Role of RA-Modulated MicroRNAs in Cancer

Multiple studies have identified RA as a potential chemoprotective and chemotherapeutic agent in various types of carcinomas due to its antiproliferative activity, exerting a growth inhibitory role at different steps of carcinogenesis. Also, RA signalling promotes cellular apoptosis, avoiding the survival of malignant cells. In fact, evidence of these inhibitory roles has been reported in breast and lung cancer, while in neuroblastoma RA signalling acts as an apoptosis-inducing factor [88]. Importantly, several microRNAs have been described as RA-modulated downstream effectors of such anti-oncogenic signals, as detailed below (Figure 3).

### 4.1. Protective Roles of RA-Mediated MiRNAs in Breast Cancer

The protective role of RA signalling in preventing breast carcinoma, mainly by promoting apoptosis and/or inhibiting proliferative and antiapoptotic signals, has been noted in the literature for some time [89,90,91,92,93]. Estrogens are considered as one of the most powerful antiapoptotic signals. On the one hand, they inhibit apoptosis and, on the other, they increase the expression of metabolic genes necessary for the growth and survival of cancer cells. RA and estrogens thus exhibit opposed roles in breast cancer modulation [94]. Interestingly, RA signalling represses the estrogen pathway via upregulation of three microRNAs (miR-210, miR-23a and miR-24-2) which in turn modulate two metabolic enzymes necessary for estrogen-promoted cellular metabolic increase in cancer. Furthermore, RA induces the expression of the pro-oncogenic miR-21, specifically in estrogen-receptor-positive cells. On the one hand, miR-21 inhibits *Maspin* expression, a pro-oncogenic factor upregulated by RA involved in anti-proliferative response, and, on the other hand, it represses the *Il1b*, *Icam-1* and *Plat* mRNA levels, respectively, thus reducing cellular motility and inflammation response during breast carcinoma progression [95]. Similarly, estrogens induce expression of two microRNA clusters, miR-17-92 and miR-424-450b, which are downregulated by RA treatment. Both microRNA clusters negatively modulate RA signalling and thus antiapoptotic factors and cancer metabolism [96]. RA signalling not only represses carcinogenesis by estrogen pathways but also reduces tumour survival by downregulating miR-130a, which, in turn, negatively modulates *Hox5a* expression, a factor that is crucial for the promotion of cellular apoptosis [97].

### 4.2. Protective Roles of RNA-Mediated MiRNAs in Colon Cancer

RA-upregulated miR-145 plays a critical role in proliferation and migration of cancer cells by reducing mRNA levels of two key antiapoptotic factors, *IRS-1* and *IGF-1R*. Colon cancer cell lines upregulate the expression of both factors, which is necessary for growth and migration of maligned cells [98,99,100,101]. Importantly, miR-145 can be considered as a protective effector in breast cancer since it negatively regulates the expression of these key factors.

### 4.3. Protective Roles of RA-Mediated MiRNAs in Neuroblastoma

RA mediates protection against neuroblastoma through different pathways, inducing cell apoptosis and inhibiting the motility and progression of metastasis through the modulation of different effectors [102,103,104]. Among these effectors, the role of microRNAs downstream RA signalling begins to stand out. To date, four RA-upregulated microRNAs that mediate protection against neuroblastoma have been described; these work by promoting cell differentiation, such as miR-432, by inhibiting cell proliferation and migration, such as miR-128, miR-10a and miR-10b, or by repressing the methylation of different key gene promoters that induce carcinogenesis, such as miR-152 and miR-340.

RA-mediated increase in neuronal marker expression is promoted by miR-432, which negatively modulates RCOR1 expression, a key repressor of neuronal genes such as GAP43. Decreasing RCOR1 is necessary for neurons to be able to differentiate, thus highlighting miR-432 as a key modulator in RA-mediated differentiation [105]. Cell mobility and invasion is considered one of the final stages of metastasis. In this context, RA mediates the upregulation of miR-128, which represses the expression of two factors involved in cell motility and in metastasis, *Reelin* and *Dcx*. Gain-of-function assays for this microRNA, and loss-of-function assays for the *Reelin* and *Dcx* genes, have shown lower cell mobility and migration, suggesting that this microRNA may exert a protective influence in cancer metastasis [106]. Before migration, malignant cells exhibit a high degree of proliferation. RA upregulated miR-10a/10b negatively modulates proliferation by repression of NCOR2, a factor that is necessary for the activation of proliferative genes. Furthermore, this microRNA negatively regulates the splicing process of pro-oncogenic factors, playing a doubly protective role in the RA signalling pathway. Finally, the expression of certain pro-oncogenic factors requires the methylation of their promoters in order for them to be expressed in malignant cells [107,108,109]. This methylation is carried out by methylases such as DNMT1 and DNMT3B. Both of these methylases are deregulated by RA administration via upregulation of miR-152, which recognises the 3′UTRs of both and represses their translation [110]. The repression of those methylases leads to expression of miR-340 which in turn represses SOX2 translation inducing differentiation of neuronal progenitors and blocking neuroblastoma [111]. Overall, these pieces of evidence demonstrate the complex interactive network of RA-regulated microRNAs in neuroblastoma.

### 4.4. Protective Roles of RA-Mediated MiRNAs in Lung Cancer

In contrast to neuroblastoma or breast cancer, the role of microRNAs upregulated by RA in lung cancer has only been poorly explored. Two microRNAs have been reported in this context, Let-7a2 and miR-512-3p. Let-7a2 negatively modulates cell proliferation, while miR-512-3p inhibits adhesion, migration and invasion of tumour cells, repressing the expression of DOCK3, which in turn promotes metastasis [112,113].

## 5. Role of lncRNAs as RA Mediated Differentiation Driver

In contrast to microRNAs, only a few cases of RA-regulated lncRNAs involved in differentiation have been reported to date. Maternally expressed gene 3 (*Meg3*) lncRNA is a pivotal mediator in the neural differentiation of amniotic epithelial cells (AECs), mediated by RA signalling. The RA/cAMP/CREB pathways activate the expression of *Meg3* in the early steps of neurogenesis. *Meg3* positively modulates neural markers, by acting as a sponge for miR-128, which in turn represses β-III tubulin, neuron-specific enolase (NSE) and polysialic acid neural cell adhesion molecule (PSA-NCAM) mRNA levels. The de-repression of these neuron-specific genes by *Meg3* promotes and increases the number of neurons, suggesting a critical role for this lncRNA in neural differentiation and nervous system development [114]. Similar to *Meg3*, *Hotairm1* is upregulated by RA in a differentiation context, i.e., granulocytic differentiation. *Hotairm1* gain- and loss-of-function assays have showed that expression of this lncRNA is critical for the gene expression program required to promote granulocytic differentiation. Curiously, knockdown of *Hotairm1* reduces RA-induced cell cycle arrest, suggesting that it is essential in antiproliferative signals mediated by RA [115,116]. 

## 6. Role of lncRNAs in an RA-Mediated Diseases

Similarly, only a few cases of lncRNAs regulated by RA have been reported in cancer, these being with regard to acute promyelocytic leukaemia (ALK), lung cancer and neuroblastoma, although there is also emerging evidence being reported under other disease conditions, such as liver fibrosis. Curiously, in one case, regulation of the lncRNA promoter activity was reported to negatively modulate RA signalling in cancer, as detailed below.

Leukocyte differentiation requires a precise balance between proliferation and cellular apoptosis. Impaired regulation of this balance leads to ALK. RA induces apoptosis to limit the number of proliferative cells and thus represses ALK. In particular, RA treatment promotes the expression of the TNF-alpha and TRAIL pathways, which in turn positively modulate the expression of caspases-3, -8 and -9, key factors in apoptosis and autophagy. Furthermore, RA induces the expression of the lncRNA *HOXA-AS2* antisense, which represses the TRAIL pathway and reduces the expression levels of caspases cited above, suggesting the presence of regulatory negative feedback modulated by RA. However, the molecular mechanism is poorly understood and requires further study [117]. Curiously, RA also induces the expression of another lncRNA, *H19*, in AKL. Upregulation of *H19* impairs complex hTERC-hTR interaction, which is necessary for telomerase function during carcinogenesis. Reactivation of telomerase is essential for oncogenic growth and survival of maligned cells in AKL [118]. 

*FOXD3-AS1* is another antisense lncRNA involved in the protective roles of RA signalling in cancer. *FOXD3-AS1* is upregulated after RA treatment and mediates crucial roles for the therapeutic effects on neuroblastoma. Upregulation of *FOXD3-AS1* reduces the tumour growth and prolongs cell survival by induction of neuronal differentiation. Mechanistically, *FOXD3-AS1* interacts with poly (ADP-ribose) polymerase 1 (*PARP1*) to repress the poly(ADP-ribosyl)ation and activation of CCCTC-binding factor (*CTCF*), leading to de-repression of downstream tumour-suppressive genes [119]. 

In contrast to the protective lncRNAs described above, RAET1K exerts its function as a cancer promoter by reducing RA signalling. *RAET1K* counteracts the protective role of RA in lung cancer, modulating the function of miR-135a-5p, which in turn represses the protein level of *CCNE1*, a cyclin required for G1–S transition. RA promotes the expression of this microRNA, arresting lung cancer cells in the G1 phase and avoiding the progression and decreasing the aggressiveness of the tumour cells. Knockdown of *RAET1K* in two tumoural cell lines, A549 and H1299, repressed *CCDE1* expression and thus hindered cell cycle progression from the G1 to the S phase. Thus, *RAEK1K* is considered as a repressor of cell arrest mediated by RA [120].

RA signalling does not always play a protective role against disease. Enhanced RA production is required for activation of hepatic stellate cells (HSCs) in liver fibrosis. In this context, *H19* increases retinol metabolism through upregulation of alcohol dehydrogenase III (ADH3), which in turn increases production of RA (Figure 1). Knockdown of *H19* or treatment with an *H19* inhibitor leads to impaired activation of HSCs and reduces liver fibrosis. Furthermore, inhibition of *ADH3* abrogates H19-mediated RA signals and HSC activation [121].

## 7. Conclusions and Future Perspectives

In this review we highlighted the pivotal role of RA-regulated non-coding RNAs as downstream effectors during cell differentiation and disease. RA signalling is a critical pathway conserved throughout evolution and it is involved in multiple biological processes, such as cell proliferation and differentiation, organogenesis, axis body establishment and homeostasis, as well as in several diseases. RA signalling has been described as a promoter signal required for cellular differentiation through its modulation of the balance between proliferation and differentiation. A finely tuned regulation of this balance is required for adequate development of many tissues and it is orchestrated by both transcriptional and post-transcriptional regulation.

Recently microRNAs have been shown to play a relevant role in the RA signalling pathway, thus establishing a new regulatory layer in this complex process. As described in this review, several microRNAs, acting as critical regulators of RA, are necessary for appropriated differentiation in several tissues. These microRNAs can either repress proliferation or promote cellular differentiation. Furthermore, RA modulates the expression of a large number of microRNAs, whose functions are currently unknown in most cases. Therefore, the role of microRNAs in the RA signalling pathway has only recently been recognised and further studies are required on this front.

Due to the fact that RA signalling has the capacity to promote cellular differentiation, it is considered as an important repressor of carcinogenesis. Additionally, multiple reports have pointed out RA signalling as a potential therapeutic tool in cancer, as RA treatment reduces growth, motility, migration and invasion of several oncogenic cell lines. Therefore, anti-cancer signals mediated by RA modulate the expression of multiple microRNAs involved in protection. Since these microRNAs repress oncogene expression, in their absence carcinoma aggressiveness is not be reduced and, consequently, RA loses its protective role.

In contrast to miRNAs, only a few cases of RA-regulated lncRNAs have been reported to date. lncRNAs display a bivalent role, acting as repressors or effectors of RA signalling. Such a bivalent function implies a complex regulation exerted by this type of RNAs and further studies are thus needed in order to deepen our knowledge of RA-regulated lncRNAs.

However, the molecular mechanism by which retinoic acid modulates the up- or downregulation of the microRNAs/lncRNAs described is still unknown in most cases. Similarly, the transcription factors that mediate the expression of these microRNAs/lncRNAs have not been studied or analysed in the RA context. Thus, it could be interesting to study the direct connections between RA, TFs and microRNAs/lncRNAs and gain more knowledge about molecular pathways modulated by RA signals.

Most microRNAs and lncRNAs described to date in RA signalling have been reported downstream of the pathway, being either activated or repressed. However, only a few cases of microRNAs upstream of this signalling pathway have been described. Given the regulatory role of these RNA molecules, it will be necessary to more extensively address the possible roles of these microRNAs in RA synthesis, transport and/or nucleus translocation. Similarly, it would be interesting to study in more depth the mechanisms and functions of each of the microRNAs involved in both the downstream and upstream RA signalling pathways. In particular, in vivo RA-regulated microRNAs should be further investigated since in vivo knockdown models would shed more light on the functional role of RA-regulated microRNAs in cell differentiation and tumour protection.

## Figures and Tables

**Figure 1 ncrna-07-00013-f001:**
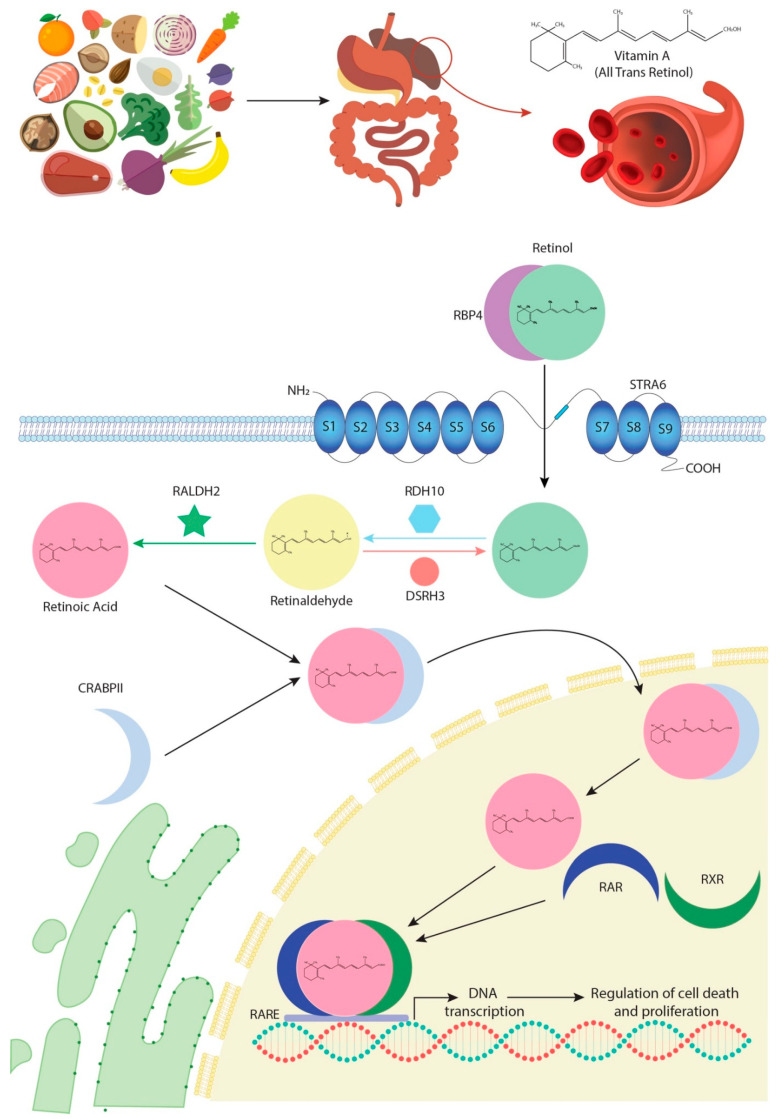
Schematic representation of the all-trans retinoic acid (RA) signalling pathways from the point it is firstly obtained from diet until it modulates gene transcription.

**Figure 2 ncrna-07-00013-f002:**
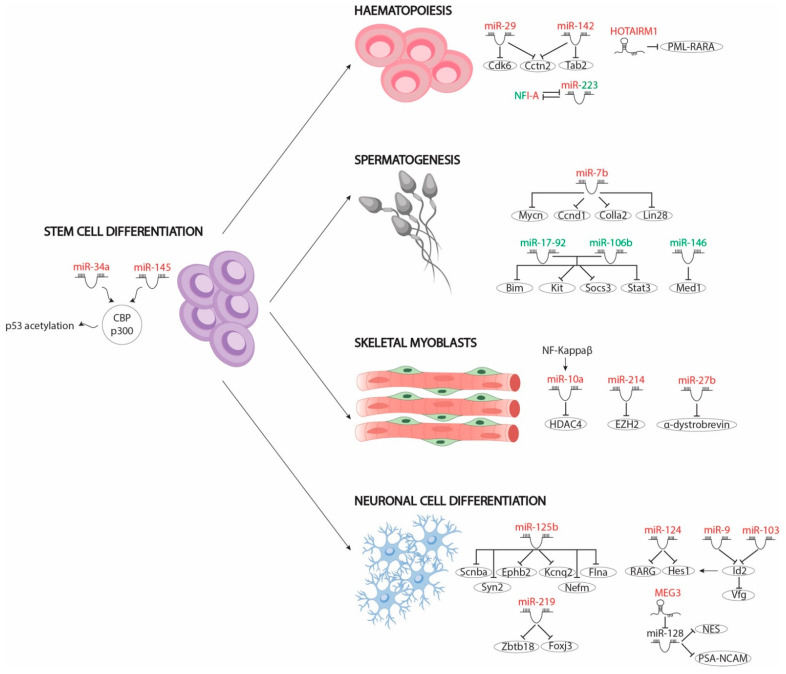
Schematic representation of the different non-coding RNAs involved in RA-regulated differentiation pathways, including stem cell differentiation, haematopoiesis, spermatogenesis and skeletal muscle and neuronal cell differentiation. MicroRNAs and lncRNAs that are upregulated after RA administration are depicted in red while those that are downregulated after RA administration are depicted in green.

**Figure 3 ncrna-07-00013-f003:**
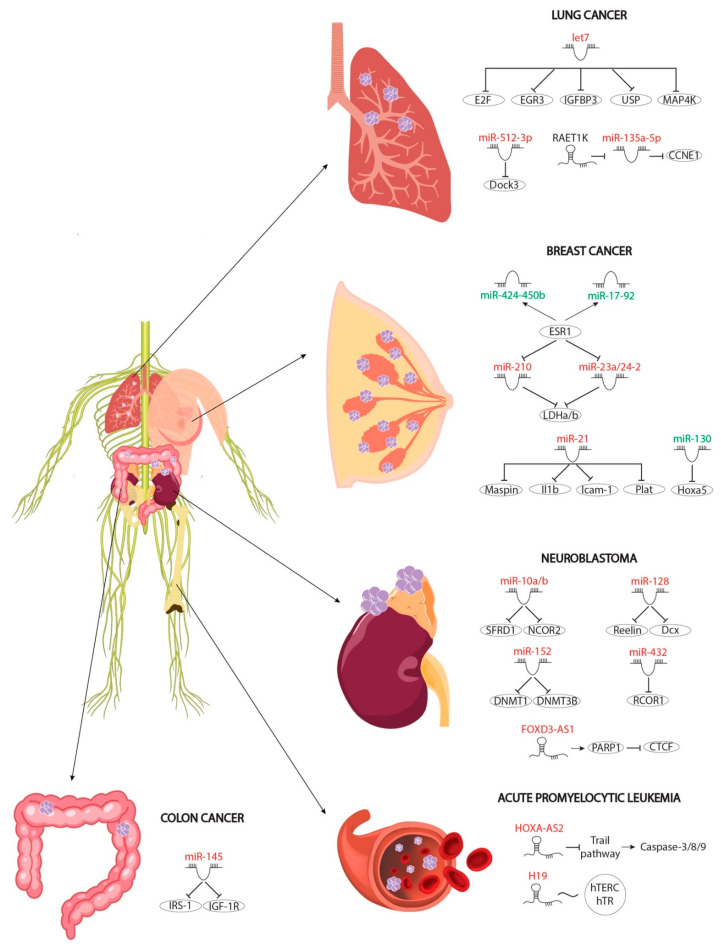
Schematic representation of the different non-coding RNAs involved in RA-regulated oncogenic pathways, including lung and breast cancer, neuroblastoma and acute promyelocytic leukaemia. MicroRNAs and lncRNAs that are upregulated after RA administration are depicted in red while those that are downregulated after RA administration are depicted in green.

## Data Availability

No new data were created or analyzed in this study. Data sharing is not applicable to this article.

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
