# Peer review of "Non-Coding RNAs in Retinoic Acid as Differentiation and Disease Drivers"

_ncrna, 2021, doi:10.3390/ncrna7010013_

Round 1

Reviewer 1 Report

The authors have substantially revised the manuscript and added figures that are helpful for the reader. Essentially, they described the role of RA signalling in downstream miRNA and lncRNA target expression. These effect have further effect on respective miRNA targets, which at the end can regulate and balance cell differentiation and proliferation. It adds an interesting overview of the topic and highlights some interesting gene regulatory circuits. Interestingly, the direct connections seems to be often not known, an aspect that could be explained a bit further and possibly briefly mentioned in the conclusion. Maybe this point could be highlighted in the discussion/conclusion a bit further. At the end, RA interacts with certain RAR that eventually leads to activation/ repression of miRNA/lncRNA target expression and experiments how to fill this gap of knowledge could be interesting. If more is known about the respective TFs that activate/repress miRNA/lncRNA expression this would be good to include (to make the fully connection).

Besides the point mentioned above, the following should be taken into consideration prior publication.

- Fig. 1: -The chosen display of retinaldehyde and retinoic acid does look more like a protein than a chemical. Instead, the authors could simply draw the chemical structure of the compounds (or a simplified version). CRABP2 binds to Retinoic Acid in the cytoplasm. In my opinion, it would be good to display where it gets released in the nucleus (converse to the association).

- Some more recent review have been published about lncRNAs (e.g. Statello L. et al. 2021, NRMCB). The authors may consider those and update references accordingly.

- Pg. 5, line 139. The statement given need better explanation. The “microRNAs are upregulated by RA signals, leading to p53 acetylation…” – how is this working? Furthermore, “acetylated p53 activates miRNA-34 expression”. This is confusing as previous sentence states that miR-34 expression by RA leads to p53 acetylation. Pls. revise so that the series of events can be better understood.

- Fig. 2 could provide an anchor point to guide through the regulatory events described in the text. This connection is not given sufficient attention and should be made more clear. For example, it is mentioned that “RA-mediated up-regulation of miR-219 is essential for repression….” (line 177). This is in the context of neuronal progenitor differentiation from ECs. However, the figure 2 displays miR-219 as being repressed during stem-cell differentiation. Thus, this is confusing and the reader gets lost as the figure is not coordinated/match with the text. Same accounts for miR-129a etc. (line 225). Or is the figure legend misleading?

- The text is better coordinated with Fig. 3. However, unclear whether the green miRNAs displayed in the figure rather refers to upregulated miRNAs (e.g. miRNA-210 related to breast cancer; line 288 is depicted in green in the figure but should be in red according to figure legend).

- Line 181: “Similarly, treatment of P19 with RA induces…” Cell lines should be introduced and explained.  In general, there are many abbreviation that have not be explained (spelled-out) first time use.

Author Response

First of all, we would like to thank the reviewer for his/her valuable and positive comments on our review.

The authors have substantially revised the manuscript and added figures that are helpful for the reader. Essentially, they described the role of RA signalling in downstream miRNA and lncRNA target expression. These effect have further effect on respective miRNA targets, which at the end can regulate and balance cell differentiation and proliferation. It adds an interesting overview of the topic and highlights some interesting gene regulatory circuits. Interestingly, the direct connections seems to be often not known, an aspect that could be explained a bit further and possibly briefly mentioned in the conclusion. Maybe this point could be highlighted in the discussion/conclusion a bit further. At the end, RA interacts with certain RAR that eventually leads to activation/ repression of miRNA/lncRNA target expression and experiments how to fill this gap of knowledge could be interesting. If more is known about the respective TFs that activate/repress miRNA/lncRNA expression this would be good to include (to make the fully connection).

Following the recommendation of the reviewer we have added a new paragraph in the conclusions of the revised version of the manuscript to highlight the point raised by the reviewer

Besides the point mentioned above, the following should be taken into consideration prior publication.

- Fig. 1: -The chosen display of retinaldehyde and retinoic acid does look more like a protein than a chemical. Instead, the authors could simply draw the chemical

structure of the compounds (or a simplified version). CRABP2 binds to Retinoic Acid in the cytoplasm. In my opinion, it would be good to display where it gets released in the nucleus (converse to the association).

Following the recommendation of the reviewer, we have modified the schematic representation of retinol retinaldehyde and retinoic acid aiming to convincing display that they are chemical compounds and not proteins. Similarly, the display of CRABP2 have been also modified as suggested by the reviewer in the revised version of Figure 1.

- Some more recent review have been published about lncRNAs (e.g. Statello L. et al. 2021, NRMCB). The authors may consider those and update references accordingly.

Following the recommendation of the reviewers, the recent review by Statello et al. (2021) have been cited in the revised version of the manuscript (ref #50)

- Pg. 5, line 139. The statement given need better explanation. The “microRNAs are upregulated by RA signals, leading to p53 acetylation…” – how is this working? Furthermore, “acetylated p53 activates miRNA-34 expression”. This is confusing as previous sentence states that miR-34 expression by RA leads to p53 acetylation. Pls. revise so that the series of events can be better understood.

Following the recommendation of the reviewer, this statement has been revised and modified in the revised version of the manuscript.

- Fig. 2 could provide an anchor point to guide through the regulatory events described in the text. This connection is not given sufficient attention and should be made more clear. For example, it is mentioned that “RA-mediated upregulation of miR-219 is essential for repression….” (line 177). This is in the context of neuronal progenitor differentiation from ECs. However, the figure 2 displays miR-219 as being repressed during stem-cell differentiation. Thus, this is confusing and the reader gets lost as the figure is not coordinated/match with the text. Same accounts for miR-129a etc. (line 225). Or is the figure legend misleading?

Following the recommendation of the reviewer, we have revised both the microRNAs/lncRNAs figure colors, representations and Figure 2 legend to provide an unequivocal color depiction in the figures, In addition, we have updated those representations to fully matched the text descriptions. We have represented in red those microRNAs that are up-regulated following RA administration and in green those microRNAs that are down-regulated following RA administration.  Figure 2 legend have been updated.

- The text is better coordinated with Fig. 3. However, unclear whether the green miRNAs displayed in the figure rather refers to upregulated miRNAs (e.g. miRNA-210 related to breast cancer; line 288 is depicted in green in the figure but should be in red according to figure legend).

In line with the previous comment, we have revised both the microRNAs/lncRNAs figure colors, representations and Figure 3 legend to provide an unequivocal color depiction in the figures. In addition, we have updated those representations to fully matched the text descriptions. We have represented in red those microRNAs that are up-regulated following RA administration and in green those microRNAs that are down-regulated following RA administration. Similarly, Figure 3 legend have been updated.

- Line 181: “Similarly, treatment of P19 with RA induces…” Cell lines should be introduced and explained. In general, there are many abbreviation that have not be explained (spelled-out) first time use.

Following the recommendation of the reviewer, the nature of P19 cell lines have been added on the revised version of the manuscript.

Reviewer 2 Report

The authors have now addressed the issues identified in the original submission and the article is suitable for publication. I really hope the authors have found the comments useful and constructive. 

Author Response

The authors have now addressed the issues identified in the original submission and the article is suitable for publication. I really hope the authors have found the comments useful and constructive.

First of all, are very happy that the reviewer considered our revised version of the manuscript suitable for publication. We would also like to thank the reviewer for his/her valuable and positive comments on our review.

This manuscript is a resubmission of an earlier submission. The following is a list of the peer review reports and author responses from that submission.

Round 1

Reviewer 1 Report

The authors have presented a comprehensive analysis of the role of ncRNA and RA assessing their biological and pathological interaction. Although the article is well structured and written, several modifications are required before publication:

Major points:

  1. Figure 1 is not clear. I suggest the authors to divide panel A and B and make two different figures: in Figure 1 to analysis the biochemistry and cell signalling of RA (e.g. include all chemical structures, provide a better picture of the cell, larger writing, degradation pathways, etc); in Figure 2, to provide a detailed analysis of ncRNA, RA and biological functions (include interacting targets for miRNA when possible).
  2. A detailed figure legend should be included with all acronyms of protein names reported in the figure.

Minor points:

169-176: section is clear but what is the link to RA?

146: capital I is i

157-158: rephrase

163-164: Thus thus.

224: keep writing Aia Ai

Spell OFT

327: delete comma

352: cell apoptosis

Reviewer 2 Report

This review compiled a number of examples that highlight the role of non-coding RNAs (miRNAs and lncRNAs) as downstream effectors during cell differentiation/proliferation mediated by retinoic acid (RA). It is a catalogue of examples that may provide an quick overview for interested readers; however, it is unfortunate that it does not go beyond being a descriptive list of events. Importantly, the actual link between RA and miRNAs i.e. how RA activates miRNA expression is mostly not given/described. While many of the given examples are known for a while and have been reviewed in the past, there is also some absence of references or being outdated. Overall, it would be more interesting to focus on recent development in the field and provide an more thoughtful outlook on the pressing questions and how those could be addressed.

The review has only one figure, which seems rather underdeveloped. More schemes on the mechanisms and overarching view of the processes and molecular players could be interesting.

Some minor points:

  1. 1A. The figure needs labelling A, B. The quality of the figure is relatively poor and sufficient and should be improved.
  2. 1 lacks a figure legend. Also Fig. 1B is not very interesting and could also go into a table. Sketching relevant mRNA targets could help. Also Meg3/Hotair has some drawings but this needs to be improved.
  3. Line 72 says: …”considered ncRNA as non-functional part of the genome”. I do not agree with this statement and it must be modified or removed. There are key ncRNAs that are highly important, such as ribosomal RNAs, tRNAs, snRNAs, snoRNAs etc. Lateron, miRNAs/siRNAs were discovered and more recently focus is given on lncRNAs.
  4. Line 77 “suggesting that ncRNA are as or more important than coding RNAs” This comparison is not appropriate. They have different functions in the cells and one can simply not compare these two groups.
  5. Line 146, typo: “human stem cell embryonic cells” > should be ‘human embryonic stem cells’.
  6. Section 3.3. This should be rewritten as it is somewhat confusing. While it is stated the RNA maintains myoblasts in undifferentiated state at the start, the paragraph ends with “RNA-induced miR-27b upregulation promotes myoblast differentiation”. This is confusing.
  7. There are many typos and mis-spellings that need correction.